# What Motivates Volunteers to Engage in Health-Related Citizen Science Initiatives? A Case Study of Our Outdoors

**DOI:** 10.3390/ijerph17196950

**Published:** 2020-09-23

**Authors:** Elizabeth Lehman, Ruth Jepson, John McAteer, Daryll Archibald

**Affiliations:** 1School of Psychology and Public Health, La Trobe University, Plenty Road and Kingsbury Drive, Bundoora, VIC 3086, Australia; darchibald001@dundee.ac.uk; 2School of Health in Social Science, University of Edinburgh, 5 Forest Hill, Edinburgh EH1 2QL, UK; ruth.jepson@ed.ac.uk; 3Independent Researcher, 23/5 Hopetoun Crescent, Edinburgh EH7 4AY, UK; john.mcateer77@gmail.com; 4School of Health Sciences, University of Dundee, 11 Airlie Place, Dundee DD1 4HJ, UK

**Keywords:** citizen science, environmental volunteering, motivations, public health

## Abstract

Citizen science is increasing in popularity but remains largely located in the disciplines of environmental and natural sciences. However, it has the potential to be a useful tool in other disciplines such as health. The aim of this study was to identify the factors for involvement (or non-involvement) in health-related citizen science projects using the Our Outdoors citizen science initiative as an example. Our Outdoors aims to understand how urban and rural shared outdoors spaces (e.g., parks, lakes, rivers, beaches) can affect human health and well-being (both positively and negatively). Understanding the motivations for involvement in such a program is likely to be useful for increasing participation rates and involvement. Qualitative research methods were used in this study in which semi-structured interviews were conducted with 12 participants from two community projects in Scotland, United Kingdom. A thematic analysis revealed five key themes pertaining to the factors that motivated engagement with health-related citizen science projects such as Our Outdoors. These include enhancing social connectedness; personal learning development; making a difference in the community; gaining health and well-being benefits; and finally, demotivating factors relating to time constraints and the term “citizen science”. This study concludes that emphasising motivating factors in the promotional material for health-related citizen science projects may increase recruitment and the active involvement of participants. Similarly, reducing the presence of demotivating factors and considering the use of the term “citizen science” is likely to encourage participation

## 1. Introduction

Citizen science refers to a participatory model of research in which non-professionals are involved as collaborators in scientific research [1]. Citizen science is becoming an increasingly popular method in a variety of research fields and contexts [2]. The emergence of citizen science has allowed researchers to undertake projects on larger scales while reducing resources and time spent collecting and analysing data [3]. Citizen science has the potential to inform and empower local policy makers and community members alike when utilised in a public health research setting [4].

The Our Outdoors initiative focuses on improving public health outcomes through engaging participants in a combination of environmental volunteering and citizen science [5]. The initiative aims to engage individuals from all socioeconomic backgrounds throughout Edinburgh and the Lothians in Scotland, to understand the impact of shared outdoor spaces on health and well-being and to understand how and why outdoor spaces affect people differently. The Our Outdoors pilot initiative began in mid-2017 and is led by the Scottish Collaboration for Public Health Research and Policy at the University of Edinburgh. 

There is mounting evidence that participation in volunteering can benefit health and well-being [6]. Due to increased physical activity, exposure to natural environments, and an enhanced sense of social connectedness, environmental volunteering may have greater health and well-being benefits compared to other forms of volunteering [7]. However, this is a developing field where more research is needed to establish conclusive evidence on the impacts of environmental volunteering on health [8]. Similarly, engaging in citizen science projects relating to public health has the potential to empower and increase the health literacy of participants [4]. However, this is also a developing area of research which requires more research to better understand how these new initiatives can impact on health outcomes. Our Outdoors is an initiative that is positioned at the nexus of these developing research areas. 

Understanding the factors that motivates engagement with citizen science may assist in optimising a project’s recruitment and retention, as well as increasing a project’s prospect of success [2,3,9]. Limited research to understand the factors that motivate engagement with citizen science has been undertaken to date. For example, in a project looking at what motivated individuals to engage with an online citizen science game, it was found that making a contribution to science and taking an interest in science were the key motivators to engagement [10]. Similarly, a study focusing on what motivates engagement with citizen cyber-science projects determined that participants were motivated by curiosity, interest in science and the desire to contribute to scientific research [11]. One further environmental conservation citizen science project determined that participants were motivated by the opportunity to contribute to scientific research, the opportunity to observe nature, and the opportunity to develop knowledge about wildlife, in this case, native bees [12]. No review-level evidence appears to exist on the factors that motivate engagement with citizen science initiatives and to date no study appears to have investigated the factors that motivate engagement with public health-focused citizen science initiatives. 

With regard to previous research on motivations to engage with environmental volunteering initiatives, this is also an under-researched area [13]. However, one study described six factors that can contribute to engagement with environmental volunteering [14]. These include contributing to the community, social interaction, personal development, learning about the environment, a general care for the environment and attachment to a particular place [14]. In addition, a study on motivations to undertake environmental volunteering among British and Austrian individuals found the factors influencing engagement were a desire to help the environment, learning experience, optimized career opportunities and esteem [15]. This study will therefore aim to contribute much-needed evidence to understanding what motivates engagement with initiatives like Our Outdoors that combine a public health focus with environmental volunteering and citizen science. 

Understanding what motivates participants in Our Outdoors may assist in encouraging initial and ongoing participation in the initiative [3]. This will allow the expectations underpinning personal motivations to be met and is likely to increase the satisfaction levels of volunteers, promoting retention and ultimately optimising the potential benefits for participants and projects alike [3]. Volunteer motivations for participation may be dynamic and may change over time [3]. As such, numerous motivations can influence a volunteer’s desire to participate in a project like *Our Outdoors.* The nature and importance of each motivation may also vary between participants and therefore contribute to a participant’s satisfaction to varying degrees [3]. Understanding more about what motivates people to participate may help to engage populations who are less likely to take part in initiatives like Our Outdoors and obtain the likely resultant health benefits. 

The Our Outdoors initiative depends on volunteers to collect data from their local outdoor areas to understand the impact of the outdoor environment on health and well-being. Therefore, understanding the motivations for volunteer involvement in a variety of population groups can enhance recruitment, facilitate high retention rates and ultimately contribute to the success of the initiative [16].

Our Outdoors had an objective in its pilot phase to identify the best methods for recruiting and retaining members of the public in the initiative. This provided the starting point for this research, which had the central aim of understanding the factors that motivate participation in the Our Outdoors initiative. With this in mind, this research sought to specifically understand firstly, what factors motivated individuals to participate in citizen science projects such as the Our Outdoors initiative, and secondly, what the implications are of the findings for the optimal design of Our Outdoors with respect to recruitment and engagement as the initiative was scaled up beyond the pilot stage.

## 2. Methods

Qualitative research methods were utilised in this study. A qualitative study design was adopted to facilitate the investigation of personal meanings, experiences and understandings, and to generate in-depth and textured data [17].

### 2.1. Sampling and Recruitment

Participants were purposively recruited for involvement based on pre-specified inclusion criteria (see Table 1). The purposive sampling of participants was a necessary measure to gain a sufficient variety of views across those involved in the Our Outdoors initiative.

Three community organisations were approached due to their previous involvement in the Our Outdoors workshops. Of these organisations, two responded with an interest in involvement in this study. Both community organisations were based within Edinburgh and the Lothians in Scotland. One of the community organisations had a focus on combatting mental health issues through encouraging social interaction and physical activity. The other organisation was a youth group located in an area of socioeconomic disadvantage.

### 2.2. Demographics

Table 2 provides a summary of the participant characteristics. A total of 12 individuals were recruited for involvement in this study in August and September 2018. All participants had been involved in some form of volunteering in the past and more than half of the participants were previously involved in the Our Outdoors initiative. The majority of the sample were retired from work. This is unsurprising given that community volunteering during retirement is a popular and well established way for retirees to remain active, both mentally and physically after leaving the workforce [18,19].

### 2.3. Data Collection

Semi-structured interviews were utilised to explore the experiences of the research participants regarding their motivations to engage in the Our Outdoors initiative. All interviews were conducted by one researcher (EL). Participants were asked a wide variety of questions relating to personal interests, volunteering experience and factors of encouragement and discouragement for participation in the Our Outdoors initiative (see Appendix A). The length of each interview ranged between 30 min and 1 h in duration and were recorded using an audio-recording device. Interviews were transcribed by one member of the research team. At the time of data collection (August/September 2018), the Our Outdoors initiative was in the pilot phase but it will eventually be rolled out across the United Kingdom.

### 2.4. Data Analysis

Thematic analysis was utilised in this study as it is an accessible and flexible method of qualitative data analysis [20]. The use of a thematic analysis allowed the themes within the data to be identified and organised to provide insight into the patterns of meaning across datasets [21]. Data saturation was achieved by interview 10, however, two further interviews had been arranged and were conducted.

The six-phase approach to thematic analysis developed by Braun and Clarke [20] guided the analysis of the collected data. Firstly, the interview data were transcribed by one researcher (E.L.) The transcripts were then read and re-read by the researcher, this allowed immersion in the collected data [22]. Secondly, codes were generated from the data and themes were established within the coded data. Thirdly, the themes were reviewed, defined and refined via a process of several group meetings, allowing the researchers to explore the relationship between themes to assist in the development of an overall “story” about the data [22]. Five overarching themes were established, as well as numerous sub-themes. These themes were richly evidenced throughout the data-sets and provided an overall depiction of the motivations related to participation in the Our Outdoors initiative. From here, a report was written in the form of a thesis in partial completion of an honours degree. Finally, this report was produced to highlight the key findings of this research. NVivo 12 software was utilised to assist in the coding and the establishing of themes within the datasets.

### 2.5. Ethical Considerations

Ethics for this research was sought from the Human Research Ethics Committee at La Trobe University, Australia (ethics reference: HEC18345). The approved ethics application was also checked by the ethics committee within the School of Health and Social Sciences at the University of Edinburgh, United Kingdom, in which written approval was granted to undertake this research.

## 3. Results

The analysis generated four themes regarding factors that motivated engagement with Our Outdoors. These were: enhancing social connectedness; personal learning development; making a difference in the community; gaining health and well-being benefits. In addition, one further theme was identified regarding demotivating factors that may discourage engagement with Our Outdoors. Figure 1 provides a visual depiction of the findings.

### 3.1. Enhancing Social Connectedness

The participants in this study felt that engagement with Our Outdoors would offer valued opportunities for social connections to be made. This was a primary motivation for initial involvement with the initiative. Opportunities to meet new people, reduce feelings of social isolation and interact and connect with new people in the natural environment were identified as sub-themes.

#### 3.1.1. Opportunities to Meet New People

Opportunities to meet new people were primary factors that influenced motivation to take part in Our Outdoors. One individual said of their previous experiences volunteering:
*“well we worked together in wee teams and from the discussion you know, ideas came into your head. I think it’s that kind of well a little bit of brainstorming, but you know you get a vibe from that level of connectedness with people…I enjoyed working with the little group we had quite a laugh doing it as well”*.—participant 8, female 60+


Another participant emphasised that participation in a project such as Our Outdoors may have the potential to encourage social interaction, and thus connectedness through providing a means to meet people with similar interests:
*“It’s kind of active and you’re…not sitting about doing nothing, you know. If it was a group, you know, meeting people … with shared interest, sharing stories, exchanging the kind of opinions and ideas and stuff … maybe they can sometimes inspire you, sometimes you inspire others”*.—Participant 12, male, 55–59


For some of the participants, involvement in citizen science projects such as Our Outdoors could provide the opportunity to meet people with “shared interests” and exchange “opinions and ideas”.

#### 3.1.2. Reducing Social Isolation

Of the participants in this study, 75 per cent were retired. For the retired participants, volunteering opportunities were often viewed as a means to maintain social connectedness and thus reduce feelings of social isolation. Participation in Our Outdoors may assist in reducing social isolation through providing an opportunity to build social connections and interact socially with other participants. One participant in retirement emphasised the crucial role that volunteering has in providing positive social environments and reducing the sense of social isolation:
*“I actually enjoy being in a public in a mixed group not in my own home and enjoy the chat that occurs cause I could just sit at home…there’s a little bit of that involved as well so I enjoy the friendship, well the new friendship. I’ve met a few people, so it’s meeting people and not being in my home I enjoy that”*.—Participant 7, female, 60+


The opportunity to engage in social activities through engagement in volunteering may be a motivating factor for retired individuals as this can reduce the impact of social isolation.

#### 3.1.3. Interacting Outdoors

It was evident that many participants associated involvement in Our Outdoors with the opportunity of social interaction in the natural environment and this appeared to be a specific motivator for engagement. For example, one female participant stated:
*“You can enjoy the outdoors, you can see the outdoors, smell it, hear it, you know—wildlife, birds and also being out in the outdoors and walking you’re more able to talk. It’s much easier to talk when you’re out in the outdoors”*.—Participant 6, female, 60+


A further participant echoed the sentiment that participation in environmental projects such as Our Outdoors can provide participants with the opportunity to engage socially with like-minded individuals amongst nature:
*“You might like meet people, like-minded people…so that’s it, maybe you can connect with other people and the environment”*.—Participant 5, female, 20–24


As such, social interaction in an outdoor setting can be a motivating factor for engagement in Our Outdoors. The opportunity to connect socially with others, reduce social isolation and interact with others in a natural environment were primary motivators for involvement in the Our Outdoors initiative.

### 3.2. Personal Learning Development

The analysis demonstrated that the participants were further motivated to engage with the Our Outdoors initiative as they viewed it as an opportunity to develop new knowledge and learn new skills. The desire to learn and the opportunity to stay mentally active were sub-themes arising throughout the participant interviews.

#### 3.2.1. Desire to Learn

The participant interviews revealed that many of the participants were motivated to engage in Our Outdoors for the opportunity to learn. For example, a male participant aged 60+ stated:
*“It would be the opportunity to learn something new, which is something I quite enjoy doing”*.—Participant 9, male, 60+


Many participants stated that engaging in the Our Outdoors initiative would encourage the transfer of knowledge between individuals with similar interests. One participant suggested that the potential for ongoing learning whilst engaging with other participants was a motivation for involvement:
*“You’re connecting with people … and learning from them. You know, you’ll learn from people…you’re going to be constantly learning”*.—Participant 8, female, 60+


#### 3.2.2. Staying Mentally Active

Many of the retried participants indicated that the opportunity to keep their mind active was a key motivating factor for their involvement in volunteering and their potential involvement in Our Outdoors. When discussing the personal benefits of participation in volunteering, one participant stated that:
*“You’ve got to keep your brain busy, haven’t you?”*.—Participant 8, female, 60+


Citizen science is a relatively new concept. As such, many individuals are unfamiliar with citizen science and how this can be utilised in volunteering projects. Despite this unknown, many participants were intrigued by this and viewed citizen science as a challenge to broaden their knowledge. For example, one male participant aged 60+ stated that Our Outdoors is:
*“…probably something that’s sort of a challenge, a bit of an intellectual challenge”*.—Participant 9, male, 60+


It is seemingly evident that for many retired individuals, participation is largely motivated by the opportunity to engage in active learning to keep the brain busy and is therefore self-directed. Citizen science provides participants with this opportunity as it is a relatively new concept to many participants, allowing them to broaden their knowledge and keep their minds active.

### 3.3. Making a Difference in the Community

All of the participants were motivated to engage with Our Outdoors due to a desire to make a difference by contributing to positive change in their communities. One participant, for example, stated that involvement in Our Outdoors would allow them to:
*“Be part of something that’s big and making a difference and leaving the planet in a better state than it is at the moment and making it better for people day to day in their lives”*.—Participant 8, female, 60+


Many participants suggested that benefitting the community provided them with a stronger motivation to participate compared with any personal gain associated with participation:
*“I think it wouldn’t be a personal gain, I think it would be a gain for everyone as a unit”*.—Participant 3, female, 60+


In addition, one participant expressed hope that participation in the Our Outdoors initiative would have a positive effect on the community’s use of outdoor spaces:
*“It would be interesting if…we saw that our data, in a follow-up, was actually used to encourage other people to use spaces or encourage the local authorities to improve them for our group so that would be quite good to see if the data had a…knock on positive effect on our community”*.—Participant 7, female, 60+


Similarly, the opportunity to engage in the Our Outdoors citizen science initiative was viewed as a potential opportunity to inform others about environmental concerns. Many participants were passionate about protecting the environment and as such were eager to advocate for change amongst their local community.
*“Well I suppose there’s the feeling that I’m contributing to some of the scientific information that’s needed in order for organisations and governments to know what happening with the wildlife”*.—Participant 6, female, 60+


The opportunity to “give back” to the community or an organisation can allow participants to also receive something in return. One participant described volunteering as being:
*“like a symbiotic relationship where you’re putting something in but you’re also getting something back”*.—Participant 8, female, 60+


As such, involvement in Our Outdoors is likely to be influenced by the opportunity it provides participants with to make a positive difference to their communities and outdoor spaces.

### 3.4. Gaining Health and Well-Being Benefits

Participants were further motivated to engage with Our Outdoors by linking their involvement in the initiative with potential improvements to both their mental health and physical health. Participants were aware that health benefits could be achieved through interaction with natural environments.

#### 3.4.1. Mental Health

It was suggested by participants that involvement in Our Outdoors would encourage outdoor activities that would increase exposure to “fresh air” and increase physical activity. Many participants associated fresh air with improved mental health outcomes:
*“…walking and doing things outdoors—I think you will change your mental attitude being out there …you benefit from the fresh air, from the surroundings around you”*.—Participant 3, female, 60+


A strong focus of the participant interviews was the established link between exposure to natural environments and the resulting mental health benefits [6,7]:
*“I suppose it spreads the word about the importance of green spaces and looking after them and the places that green space has in psychological wellbeing…just feeling more at ease, relaxed and joyful about the wildlife, the birds and obviously the feeling that you have with the exercise, regular exercise”*.—Participant 6, female, 60+


Our Outdoors was viewed by many participants as an opportunity to improve mental and physical outcomes for both themselves and the wider community through interaction with the natural environment and engagement in increased levels of physical activity.

#### 3.4.2. Physical Activity and Outdoor Space

The opportunity to be active in outdoor spaces was seen to be a motivating factor for participation in the Our Outdoors initiative. One participant spoke of the outdoors being a direct motivation for involvement in Our Outdoors*:*
*“Getting outdoors is also an encouragement you know, the word ‘outdoors’ is good in terms of wanting to get involved”*.—Participant 2, male, 60+


Some participants suggested that Our Outdoors would satisfy a need to engage in outdoor activity and linked spending time outdoors with positive health outcomes.
*“Well you assume that the more people that get out into the fresh air and seeing the attractive environment, that will make them feel better psychologically and physically fitter”*.—Participant 6, female, 60+


As such, participants made the presumption that involvement in Our Outdoors would encourage them to interact with the natural environment and provide them with the potential to improve overall health and well-being outcomes.

### 3.5. Demotivating Factors

Despite overwhelmingly positive feedback on the Our Outdoors initiative, it was evident that there are a variety of factors that may discourage participation.

Unsurprisingly, time commitment was a significant demotivating factor for several of the participants. One participant stated that:
*“the only thing that would discourage me would be you know the organisation asking for a very regular commitment”*.—Participant 6, female, 60+


This was a recurring notion presented in the participant interviews, with many of the participants dedicating their time to more than one volunteer project.

Participants were also concerned that a project such as Our Outdoors would require participation in difficult or complicated tasks outside of their abilities:
*“…if it came with a lot of information of descriptions of what the project was about, if it was really lengthy or if it seemed that the tasks were a little too complicated…that would be a discouragement or if it seemed it was going to be online but [took] too long to complete … that might be a discouragement too”*.—Participant 7, female, 60+


In addition to this, some participants were concerned about “letting down” the project team due to an inability to manage what may be expected of them:
*“I think if they want me [to participate] and I can’t manage it, you feel as though you’re letting them down”*.—Participant 1, female, 60+


As stated earlier, the information collected from the Our Outdoors initiative is proposed to be distributed to local governments and councils to stimulate improvement in local outdoor spaces. However, many participants did not believe that local councils would be motivated to use the data collected and the recommendations provided from Our Outdoors to improve the local environment.
*“What is the purpose, is it looking for improvements in outdoors spaces?… cynically I would say, ‘well if I’m going to [be involved] is the council going to be interested?’”*.—Participant 2, male, 60+


Many participants were aware that achieving societal changes can be difficult. This can act as a demotivating factor for many participants due to the lack of the motivational factors in “making a difference”.
*“A lot of things that happen though is social problems it’s the problems that we have socially the way people you know, people are disenfranchised, disengaged you know … not everybody has the same attitude as me … so I hate that attitude so maybe, maybe that would sort of, I don’t know, sometimes people’s attitude discourages me”*.—Participant 12, male, 55–59


#### The Term “Citizen Science”

To recap, citizen science involves engaging members of the general population in the data collection and analysis processes of research projects [9]. Although citizen science has been utilised in a variety of fields of research for many years, it is a relatively new concept in public health [2] to people generally. When asked their initial thoughts on the concept of citizen science, it was evident that participants often did not understand the meaning of the term. Prior to participating in the interviews, two of the participants had researched the meaning of citizen science due to a limited understanding of the term. Many of the other participants found the term somewhat confusing and poorly phrased.
*“Well it doesn’t describe it initially, you know I’m sure a few people thought ‘what the heck does that mean?’”*.—Participant 1, female, 60+


Some participants noted that the term “science” may be confusing and prompt the generalised notion of science only occurring in a “laboratory”. Other participants noted concern with the term “citizen”. It was suggested that the word “citizen” may be considered unwelcoming by prospective participants:
*“It sounds [like] a slightly stand-offish term because the word citizen doesn’t sound very welcoming … I wouldn’t identify with the word citizen”*.—Participant 7, female, 60+


One participant suggested that renaming citizen science to:
*“people’s science [because] I don’t know citizen seems less connected”*.—Participant 8, female, 60+


Many of the participants did not have a strong understanding of citizen science and their misinterpretations of this term acted as a demotivating factor for involvement in the Our Outdoors initiative.

## 4. Discussion

This research sought to understand what factors motivated individuals to participate in the Our Outdoors initiative and to assess the implications for the optimal design of Our Outdoors with respect to recruitment and retention. This section will discuss the findings with respect to these two aims, as well as the implications for future research.

### 4.1. What Factors Motivate Individuals to Participate in the Our Outdoors Initiative?

Understanding participant motivations for engagement in Our Outdoors may facilitate high levels of participant satisfaction, increasing the recruitment and retention of participants, and therefore improving the overall success of the initiative [2,4,5]. The themes generated from the analysis of the participant interviews related to social connectedness, personal learning development, making a difference in the community, and the potential of health and well-being benefits through participation in Our Outdoors. Importantly, many participants also identified numerous factors that are likely to act as demotivating factors for engagement with Our Outdoors.

Social connectedness has been noted as a motivating factor for involvement in volunteering in previous studies [23,24] and the findings here align with that literature as social connectedness appeared to be the strongest motivating factor for participation in Our Outdoors. As stated previously, volunteering during retirement is an extremely popular activity that can provide retirees with mental and physical stimulation after leaving the workforce [18,19]. The popularity of volunteering for retirees is likely why the vast majority of the participants in this study were over the age of 60 and retired. Retired individuals are more likely to experience social isolation [25]. Much of this population group suggested that participation in volunteering projects like Our Outdoors was motivated by a desire to maintain social connectedness and reduce the impact of social isolation throughout retirement. These findings contribute to existing research that suggests that the opportunity for social interaction through participation in volunteering projects, similar to Our Outdoors, can be a motivating factor for initial and ongoing participation [11,23,24].

There is considerable evidence to suggest that involvement in volunteering projects can have positive impacts on health and well-being [6,7,8], however the potential health benefits of participation is rarely mentioned as a motivation to participate in previous literature [9]. The findings of this study, however, clearly demonstrate that participants were motivated to engage in the Our Outdoors initiative due to its potential for health benefits. Many participants suggested that involvement in Our Outdoors would increase their exposure to fresh air and improve their psychological well-being. This finding is significant as it provides evidence to suggest that the apparent health and well-being benefits of participation in environmental volunteering projects, such *as Our Outdoors*, are motivating factors for participation.

This study also established that personal learning and development was a key motivating factor for participation. Participation in Our Outdoors was motivated by the opportunity to learn about the local natural environment and its existing issues, such as litter problems and poorly lit public spaces. This finding largely correlates with previous studies that have found that learning was a factor that can increase the strength and quality of motivation for volunteer participation [11,16,23]. As stated previously, many of the participants in this study were retired and for this population group, participation in volunteering was strongly motivated by the opportunity to engage in learning to keep the brain active. These findings contribute to existing literature suggesting that learning opportunities are a key motivating factor for participation in environmental volunteering projects [11,16,23].

The findings of this study also suggest that participation in volunteering projects such as Our Outdoors is strongly influenced by the desire to make a difference or to contribute in some way. In particular, the opportunity that Our Outdoors would provide participants with to make a positive impact on their local communities was a key motivating factor for participation. The desire to contribute positively has been found to be a motivation for participation in citizen science projects in multiple studies [3,12]. This was strongly emphasised in the participant interviews in which participation in Our Outdoors was more strongly motivated by the potential for positive outcomes in the community and local outdoor spaces. This emphasises the notion that participation in projects like Our Outdoors can be strongly motivated by making a difference and contributing to positive change in the natural environment [3,12,25].

Despite the numerous motivating factors for involvement in the Our Outdoors initiative, there are also apparent demotivating factors. A significant finding of this research was that use of the term “citizen science” may act as a demotivating factor. There is, to our knowledge, no literature to suggest that the term “citizen science” may be a demotivating factor and as such, this is a new finding presented in this research. The participant interviews demonstrated that the term “citizen science” can be misunderstood or misinterpreted. Many participants found the concept and phrasing of citizen science to be confusing and unrelatable. Some participants also suggested that the term “citizen” was uninviting and unrelatable and therefore acted as a demotivating factor for participation in Our Outdoors. This is a significant finding within this research, as there is very limited literature to suggest that the use of the term “citizen science” may negatively impact engagement with projects similar to Our Outdoors. In addition to this, it is interesting to note that Participant 8 suggested that a more inclusive alternative to the term “citizen science” would be “people’s science”. The use value of this alternative term for promoting volunteer engagement in projects like Our Outdoors could be further explored. For example, future research could be undertaken to assess the levels of volunteer engagement of similar projects, where one is referred to as a “citizen science” project and the other as a “people’s science” project.

The most significant demotivating factor for participants in this study was time. This is similar to numerous studies investigating general volunteering in which time constraints or a perceived lack of time was the most significant demotivating factor [12,16,24]. Another significant demotivating factor was a concern that the project would not achieve the necessary results, or the change that was initially intended. This was based on the idea that making changes to the local natural environment would be difficult and local governments would not take the findings and suggestions of the Our Outdoors initiative seriously. This aligns with the findings of numerous studies suggesting that the opportunity to make a difference or inspire change is a key motivating factor [3,12,25]. The absence of this motivating factor can therefore act as a demotivating factor [16].

### 4.2. What Are the Implications of the Findings for the Optimal Design of the Our Outdoors Initiative?

To recap, understanding the motivations and meeting the expectations of involvement can encourage higher levels of participant satisfaction and this can in turn increase the recruitment and retention rates of participants in a project [3]. Increased retention or participation rates throughout a volunteering project are crucial to the overall success of a project as long-term participation allows participants to develop an increased understanding of their role, increasing their ability to perform [11].

Based on the above, it is crucial for the Our Outdoors initiative to plan with an understanding of the importance of volunteer motivations throughout the course of the project. Incorporating (and crucially, *highlighting*) opportunities for social interaction, personal learning development and making a difference in the community may encourage increased satisfaction and retention rates during the roll-out of the Our Outdoors initiative throughout the United Kingdom [3]. In particular, emphasising the opportunities for social connectedness through participation in the Our Outdoors initiative in any promotional material may motivate more individuals to participate. The findings in this study suggest that many participants were also motivated to engage in the Our Outdoors initiative due to the potential health and well-being benefits of participation in environmental volunteering projects [6,7]. As such, promoting and addressing the potential health and well-being benefits of participation in recruitment and training materials may increase potential participant engagement in the project.

It is also crucial to consider the demotivating factors for participation. A key finding of this study related to the use of the term “citizen science” acting as a demotivating factor for participation. Although two of the participants had researched the term to gain a deeper understanding, all other participants found “citizen science” to discourage their participation in Our Outdoors. The insight that this study provides into participant thoughts on the term “citizen science” indicates that the Our Outdoors initiative may need to consider the use of an alternative term to citizen science in order to potentially increase recruitment and retention rates. Alternatively, an explanation of the term citizen science could be provided to potential participants in promotional materials. In addition, the findings and recommendations of this study have the potential to be adopted by public entities endeavouring to engage participants in projects not unlike citizen science.

### 4.3. Implications for Future Research

Further research is required to understand more regarding the motivations for participation throughout all phases of citizen science projects, from data collection to advocacy. Motivations for participation are dynamic and often change over time; the motivations for initial involvement are likely to differ from the motivations for long-term or ongoing involvement in a project [13]. This study focused on motivations for initial engagement with the Our Outdoors initiative and provides a basis to assist in the initial recruitment of participants in Our Outdoors. It is suggested that longitudinal research with a similar nature to this study should be conducted to understand how motivations for participation change throughout all phases of initiatives similar to the Our Outdoors initiative.

## 5. Conclusions

The primary aim of this study was to understand what motivates engagement with citizen science projects such as the Our Outdoors initiative. Citizen science projects which focus on health present a unique opportunity for participants. Participants have the potential to simultaneously contribute to their local communities in a positive way and gain health benefits from participating. The findings of this research will allow the organisers of similar citizen science projects to plan the design projects in order to optimise recruitment and to continue meaningful engagement. This will help ensure that as many people as possible can gain the likely health benefits that involvement in such projects can provide.

## Figures and Tables

**Figure 1 ijerph-17-06950-f001:**
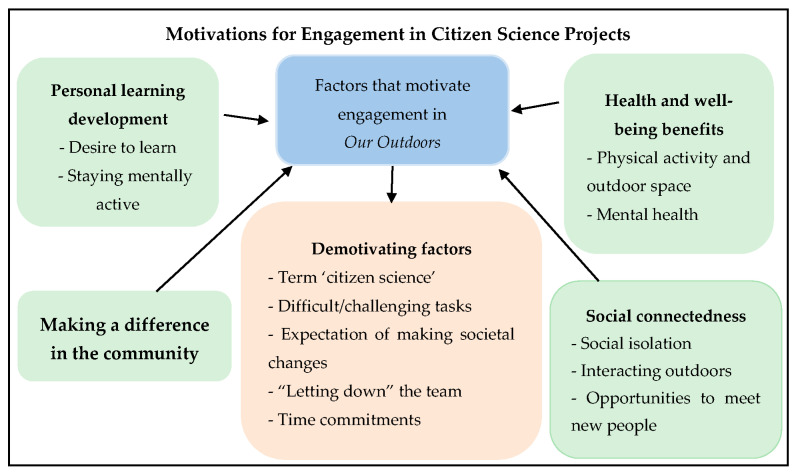
Motivations for Engagement in Citizen Science Projects—visual depiction of the findings.

**Table 1 ijerph-17-06950-t001:** Inclusion Criteria.

Inclusion Criteria
1. An expressed interest for future involvement in the Our Outdoors initiative
2. Previous involvement in an Our Outdoors workshop or intent to participate in the future
3. Possession of the ability to provide consent
4. Over 18 years of age

**Table 2 ijerph-17-06950-t002:** Participant Demographics.

Participant No.	Age Group	Gender	Employment Status	Volunteered Before (Y/N)	Previously Involved in Our Outdoors (Y/N)
Participant 1	60+	Female	Retired	Y	Y
Participant 2	60+	Male	Retired	Y	N
Participant 3	60+	Female	Retired	Y	N
Participant 4	60+	Male	Retired	Y	N
Participant 5	20–24	Female	Student	Y	Y
Participant 6	60+	Female	Retired	Y	Y
Participant 7	60+	Female	Retired	Y	Y
Participant 8	60+	Female	Retired	Y	Y
Participant 9	60+	Male	Retired	Y	N
Participant 10	60+	Female	Retired	Y	N
Participant 11	40–44	Female	Community Development	Y	Y
Participant 12	55–59	Male	Youth Club Manager	Y	Y

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
