# Peer review of "What Motivates Volunteers to Engage in Health-Related Citizen Science Initiatives? A Case Study of Our Outdoors"

_ijerph, 2020, doi:10.3390/ijerph17196950_

Round 1
Reviewer 1 Report
Aim of the work is to identify the factors for involvement (or non-involvement) in health-related citizen science projects using the “Our Outdoors” initiative as an example. The theme is extremely interesting due to the spread of health-related citizen science participation in initiatives aimed at improving the well-being of cities and urban spaces, but contents are not sufficiently treated in an in-depth way for the journal. The analysis of results may be much more analytical and commented in a critic way, such as a research work should be: maybe the results could be analyzed also from the point of view of usefulness of a public entity which could use it to activate similar programs. Furthermore, literature review and state of the art section is missing, no exam is done for providing an adequate framework regarding the level of knowledge on the topic and in what way is treated by other Authors worldwide or in the country. The basis of a scientific research work are missing, therefore the paper is not recommended for publishing.
Reviewer 2 Report
The sample size is very small (12) and also age is very high.
How many people gives what are the basic QUESTIONNAIRES IN TABLE forms .
How such a old age volunteers were active? how this is effective work?
How many days are volunteered? This is very weak paper and must improve. Although they talk about mental health , the age factor is more important. Volunteer itself more aged, so how they can help community.
my other concern:
Even results of survey should be in the tabular form for easy understanding .
How one can confirmed such a old age volunteers as their psychological thinking compare to student whose age is less.
Reviewer 3 Report
Thank you for the opportunity to review this very interesting paper which is clear, concise and very readable. I consider this to be a timely and relevant topic and the motivations of people to partake in alternative forms of science are of increasing importance at the moment! Overall, this as a report is well considered and useful in a resticted subject area but also has much wider relevance too.
Key Comments;
Abstract = appropriate, clear and concise. A good precis.
Introduction = sets scene clearly. Demonstrates study background and value/benefits well. Good, elarly reporting of an ongoing initiative. Line 59 - interesting point about the value of "natural"/environmental volunteering on health and well-being; consider reviewing some of the work looking at volunteering with animals, biophilia links and animal-assisted activities perhaps? Clear justification of study.
Methods = clear. ethical review/consideration noted and acknowledged. Clear qualitative design with clear inclusion/exclusion criteria. Section 2.3, line 115/116 - were all questions standardised and asked in the same order? Are questions worthy of inclusion for full reporting? Was the same interviewer used?
Results = well structured and generally very clear and reabable. Line 147, suggest remove term "below" in referring to figure 1 (as figure is not actually below!) Figure 1 after line 147 - some text is missing/obscured; suggest review and amend for full clarity and increase font size for easy access. Consider colour use to ensure full accessibility also. However, figure is a lovely visual addition. Line 230-232, line spacing is inconsistent. Line 344 - i love the term "peoples science" - would love to have seen consideration of this further in discussion.
Discussion = clear and well structured. Nice initial recap/scene setting. Good points, well considered with good logical flow. Line 389-400 - referencing style changes from numerical to Harvard style! Very interesting discussion point line 401-411 - wide ramifications! Line 451 - suggest "overtime" should be amended to "over time".
Conclusions = appropriate for paper.
Overall, i really liked this study and this paper - well written, a pleasure to read and review and a very interesting topic/subject area. Thank you!
Round 2
Reviewer 1 Report
The efforts made by the Authors for improving the manuscript is well appreciated. By adding the appendix A the results are already much more useful and this implies a compliment to the authors who have worked to make the paper accessible to a wider scientific audience. Concerning the literature review section, it would be better if can be added an analysis of similar scientific studies about the factors that affect the involvement (or non-involvement) in other health-related citizen science projects worldwide. This scientific study would allow the work to be valid on a consolidated scientific basis, highlighting the knowledge of the subject and mastering the fundamental aspects. Furthermore, if possible, revise the content of the work by avoiding unnecessary repetition of concepts in the Introduction section regarding the importance of such type of initiatives and so on. For the reasons set out, major revisions are required before the publication.
Reviewer 2 Report
Though authors have addressed almost all my comments
Author Response
Thank you for your comments.